# Using a Stochastic Agent Model to Optimize Performance in Divergent Interest Tacit Coordination Games

**DOI:** 10.3390/s20247026

**Published:** 2020-12-08

**Authors:** Dor Mizrahi, Inon Zuckerman, Ilan Laufer

**Affiliations:** 1Department of Industrial Engineering and Management, Ariel University, Ariel 40700, Israel; inonzu@ariel.ac.il (I.Z.); ilanl@ariel.ac.il (I.L.); 2Data Science and Artificial Intelligence Research Center, Ariel University, Ariel 40700, Israel

**Keywords:** cognitive modeling, decision-making, tacit coordination, divergent interest, autonomous agent, social value orientation (SVO)

## Abstract

In recent years collaborative robots have become major market drivers in industry 5.0, which aims to incorporate them alongside humans in a wide array of settings ranging from welding to rehabilitation. Improving human–machine collaboration entails using computational algorithms that will save processing as well as communication cost. In this study we have constructed an agent that can choose when to cooperate using an optimal strategy. The agent was designed to operate in the context of divergent interest tacit coordination games in which communication between the players is not possible and the payoff is not symmetric. The agent’s model was based on a behavioral model that can predict the probability of a player converging on prominent solutions with salient features (e.g., focal points) based on the player’s Social Value Orientation (SVO) and the specific game features. The SVO theory pertains to the preferences of decision makers when allocating joint resources between themselves and another player in the context of behavioral game theory. The agent selected stochastically between one of two possible policies, a greedy or a cooperative policy, based on the probability of a player to converge on a focal point. The distribution of the number of points obtained by the autonomous agent incorporating the SVO in the model was better than the results obtained by the human players who played against each other (i.e., the distribution associated with the agent had a higher mean value). Moreover, the distribution of points gained by the agent was better than any of the separate strategies the agent could choose from, namely, always choosing a greedy or a focal point solution. To the best of our knowledge, this is the first attempt to construct an intelligent agent that maximizes its utility by incorporating the belief system of the player in the context of tacit bargaining. This reward-maximizing strategy selection process based on the SVO can also be potentially applied in other human–machine contexts, including multiagent systems.

## 1. Introduction

Previous research has shown that it is possible to aid an artificial agent in predicting human decisions in the context of coordination problems by incorporating prominent solutions, known as focal points, into learning algorithms [1,2,3,4]. It was also found [4] that classifiers achieved 40 to 80% higher correct classification when predicting the choices of human players when a focal point learning approach was used in comparison to regular classifiers. Focal point methods were demonstrated to be useful in communication-impoverished contexts [3,5]. 

However, integrating cognitive models into agent models is not only useful in communication-impoverished settings but also in order to minimize the interaction between the artificial agent and the human partner or among teams of automated agents. Ref. [6] suggested to incorporate into the human–agent collaborative architecture two important elements: knowledge of the human strategy and the execution of a plan that considers the human partner. Furthermore, the framework suggested by [6] was demonstrated by an example taken from rehabilitation robotics. By using this example, the authors demonstrate two main principles of artificial agent behavior: adaptability, that is, the ability of the agent to adapt to human behavior, and reliance on a repository containing human-centric strategies. According to [6], these principles may improve the prediction intent of the agent and can therefore diminish the cost of conflict between the user input and the predicted action by the intelligent agent. 

It has already been suggested that achieving successful human–machine collaboration requires the modeling of human behavior for predicting people’s decisions [7,8]. It has been shown, for example, that information about drivers (i.e., driving style) improves the prediction models regarding the use of an automated assistive system [9]. This improved prediction is important since it can reduce the amount of communication between drivers and automated systems that will save computational cost. In the same vein, it has been demonstrated that behavioral economic models of people incorporated into computational approaches enhance the efficacy of advice provision strategies [10].

Likewise, when considering multiagent systems (MASs), in the field of teammate modeling, agents learn about other agents and construct models of them (e.g., as either competitive or cooperative) in order to better anticipate their behavior to achieve a more effective collaboration with them. This task is complicated since agents exemplify a wide range of behaviors depending on the context (see a review in [11]). Thus, combining behavioral models with computational approaches is also important for improving the efficiency of MAS design frameworks, especially since these may also assist humans in contexts such as smart house automation that incorporate various intelligent sensors [12]. 

In this study we have utilized the framework of behavioral game theory to construct an agent that can choose an optimal playing strategy based on the player’s motivations embedded within the social value orientation (SVO) theory. The SVO theory pertains to the preferences of decision makers when allocating joint resources between themselves and another player in the context of behavioral game theory [13,14]. Specifically, we have constructed an autonomous agent whose aim was to achieve an optimal solution by maximizing the payoff gained across a set of tacit bargaining games (e.g., [15,16,17]). The agent selected stochastically between one of two possible solutions—a greedy solution or a focal point solution—and its decision was based on assessing the probability of a player converging on a focal point using a model that incorporated the SVO of the player as well as several game features. It should be noted, however, that in the context of the current study, prosocial behavior cannot be distinguished from inequity aversion [18] since both concepts can explain the tendency towards cooperative behavior. Nevertheless, the purpose of the current study was not to distinguish between the motivations that lead to different resource allocations but rather to utilize the social preference expressed within the SVO framework as part of the behavioral model used by the agent. 

We have chosen to examine the behavior of the autonomous agent in the context of tacit bargaining for two reasons. First, bargaining solutions are applicable in resource-sharing systems such as human–machine networks (HMNs) where players trade off between maximizing their own utility and the losses of others [19,20]. Second, as mentioned above, in many contexts where human–artificial agent coordination is needed and is not necessarily performed tacitly, communication between the artificial agent and the human is costly in terms of both time and processing load. Therefore, evaluating the agent’s model in the context of the most basic form of bargaining without any communication could be important for scenarios in which it is needed to minimize the communication cost. Relying on an efficient agent model will enable the agent to predict when it is worthwhile to cooperate or share resources and alternatively when it is more advisable to act selfishly or independently of the other teammates. The development of an effective model in bargaining scenarios is therefore also important for MAS design frameworks [21,22]. To the best of our knowledge, this is the first attempt to construct an intelligent agent that maximizes its utility by incorporating the belief system of the player in the context of tacit bargaining.

## 2. Methods

To model the strategic behavior of players in the context of divergent interest tacit coordination in this study, we have utilized the “Bargaining Table” task [15,17]. The game utilized in the current study is described in detail elsewhere [23] and here we describe it only briefly. The Bargaining Table poses a tacit coordination task since the two players must assign each of the discs scattered on a 9 × 9 game board to one of two squares tacitly, without knowing what assignments were performed by the other player. Each of the squares (e.g., one blue, the other orange) represents one of the players. In this setting, the value of each disc is given as payoff only if both players assign it to the same specific player (blue or orange square). If a given disc was assigned differently by the players (i.e., each player assigned the disc to a different square), both players received a penalty of 20% of the value of the disc. The behavioral results of the Bargaining Table game enabled us to model the players’ behavior, while examining the tradeoff between selecting a prominent focal point solution and a more self-maximizing one. The examination of the players’ behavior was done while considering the SVO value [13,14] of each of the different players. The SVO value helped us quantify the motivation behind the players’ behavior (e.g., [24,25,26]) and thus better explain the behavioral results. 

### 2.1. Participants

For the construction of the behavioral model, we gathered data from 38 university students that were enrolled in one of the courses on campus (13 female participants; mean age = ~25, SD = 4.02). To evaluate the performance of the automated agent, 93 university students that were enrolled in one of the courses on campus (42 females; mean age = ~24, SD = 3.45) were recruited. The study was approved by the IRB committee of Ariel University. All participants provided written informed consent for the experiment.

### 2.2. Procedure

To construct the behavioral model, we gathered data from players who played a series of 10 random game boards since we aimed to construct a generic model as much as possible. The following parameters were randomized by using a uniform distribution: the number of discs (~U(1,8)); the numeric value of each disc (~U(1,5)); and disc position. However, for estimating the performance of the autonomous agent, the data were gathered from 10 predefined game boards so that we would be able to compare the actual distribution of the players’ results versus the results obtained by the autonomous agent (for more details on the structure of the predefined game boards, see Figure A1 and Table A1 in Appendix A). Each player played four games in which they were in a dominant position (i.e., X_3_ = “1”, see details in Section 2.3), four games in which they were in a weakness position (i.e., X_3_ = “-1”), and two games in which the position of the player was that of equality (i.e., X_3_ = “0”). It is important to note that the players were not informed of the results of the previous iterations while playing the 10 board games. Thus, each player encountered a series of one-shot games and therefore a reciprocal behavior could not have emerged as in a repeated game scheme where dynamic changes in the players’ strategy are expected to occur. Since in our model we were interested in utilizing a stable social orientation measure, a repeated game scheme was not suitable for the current study. The complete set of games, which includes all 10 predefined games, was designed in such a way that if the player chooses focal point solutions in all 10 games, the total points they receive is 52 for both the “blue” player and the “orange” player. An illustration of all the game boards in the predefined set, together with a table containing the characterization of all the game parameters (as defined in Section 2.3) can be found in Appendix A and in [23]. 

Next, the SVO of each player was measured using the “slider” method [13]. Each player was presented with a slider method application and answered six resource allocation tasks by using the slider. The slider method allows computing an SVO angle for each of the participants that reflects their proximity to one of the known social orientation categories such as competitiveness, individualism, cooperativeness, altruism, and aggression. For more details about the computation of this measure, see ref. [13]. 

### 2.3. Game Board Parameters and Model Construction 

In order to model the players’ behavior and evaluate the probability of implementing a focal point, we utilized the following six parameters (denoted as X_1_ to X_6_ in the model, respectively; see Figure 1) as follows. First, we have devised a novel parameter, namely, the Focal Point Prominence (FPP) measure (X_1_). The FPP value is calculated based on Euclidean distance and reflects the level of focality (e.g., [27,28,29]) of each board location. This measure was included in the model since it has been shown that the most prominent strategy players utilize in order to converge on a focal point is the rule of closeness by which the player assigns each disc to the closest available square.

Thus, the FPP indicates the saliency of a specific focal point by implementing the closeness rule considering a single disc on the game board. Specifically, this measure quantifies the ratio between the maximum Euclidean distance to the minimum distance of a single disc on the board (marked by d1) with respect to each of two points, each designating one of the players. The minimum value is 1, which represents the case when there is no focal point, and the maximum value is 7 in the case when the saliency of the focal point entails the maximal probability to implement a focal point solution. In the latter case it means that the disc is in the closest possible proximity to one of the players. The index is calculated as follows:(1)$$FPP=\;\frac{Max\;\left(D\left(r1,d\right),D\left(r2,d\right)\right)}{Min\;\left(D\left(r1,d\right),D\left(r2,d\right)\right)}$$
where:(2)$$D\left(r,d\right)=\;\sqrt{{\left(r.x-d.x\right)}^{2}+{\left(r.y-d.y\right)}^{2}}$$

The next two parameters were based on Isoni et al. [15,17,30]: the first one is the Expected Revenue Proportion (ERP) (X_2_), which reflects the percentage of points that the player receives if both players implement a focal point solution according to the closeness rule. We have normalized this value to fit the [0, 1] range, where 0 denotes that the player is not expected to receive any point, and 1 denotes the case in which the player is expected to get all the points in the game, if a focal point solution is implemented. An additional parameter based on the ERP is the player’s strategic position (X_3_), which is defined by a feature with three states with the assumption that a focal point solution is implemented in each: “1”—when the player gains over 50% of the point. In this case the player is in a *dominant* position; “0”—when the player gains exactly 50% of the points. In this case the points are divided *equally* between the players; and “−1”—when the player gains less than 50% of the points and is in a *weak* position. It should be emphasized that the points gained by each of the players are distributed only after the players have completed all 10 games. To get a better intuition regarding the strategic position parameter (X3), see Appendix B.

Additionally, we have defined two basic game features: the total number of points (X_4_) and the total number of discs (X_5_) on the game board. The last and sixth parameter was the SVO angle (X_6_) of the player.

Based on the above six parameters (X_1_–X_6_) associated with each game instance, we have constructed a model to estimate the probability of a player to implement a focal point solution. The output parameter, Y, receives either a value of 1 (true) or 0 (false) depending on whether a focal point solution was chosen or not. Since our results are binary (Y = 1 or Y = 0), and our goal was to produce a probabilistic measure, the most suitable classification method for our problem belongs to the family of ensemble classifiers [31,32]. We used the bootstrap aggregated ensemble of complex decision trees (e.g., [33,34]), which provided the best results among different ensemble classifiers.

In this model, there are two hyper-parameters: the number of trees and the maximal tree depth. First, the maximum tree depth was limited to prevent overfitting. Next, to determine the best model configuration, we have used cross-validation with five folds, each containing 76 different random observations (a total of 380 observations). The values for tree depth ranged between 1 and 50 and the number of trees ranged between 1 and 300. In order to validate the optimal model configuration, we switched again into a classification problem, where the classification threshold was set to 50%. The total prediction accuracy of the model that included the SVO feature was 85% (Table 1). In contrast, excluding the SVO feature from the model resulted in a dramatic decrease in prediction accuracy to the value of 67.89% (Table 2). Furthermore, we can also observe a drastic decrease in the positive predictive value of class 1 (choosing a focal point solution) when the model is without the SVO feature (Table 2).

### 2.4. Optimizing an Agent Model

In this section, our goal was to use the previously described model to build an autonomous agent that achieves better results than human players in tacit coordination. To that end, two agent subtypes were created: greedy and focal.

Importantly, the agent was trained only on the data obtained from the random game boards (see Section 2.2) included in the first part of the study and not on any of the data obtained in the second part of the experiment including the fixed set of games (Section 2.2 and Appendix A). This separation into training and test datasets was done to avoid model overfitting. The agent’s goal was to maximize the average number of points gained across all games with as minimal a variance as possible. The agent’s performance was compared to the results obtained by the human players and the two subagents (greedy and focal) were compared to each other.

#### 2.4.1. Modeling Deterministic Agents

In this section, we will describe the structure of the various deterministic agents that were designed to maximize their earnings in coordination games and the rationale behind the chosen strategies. First, we wanted to examine the results of two classes of static agents that were designed to play a specific strategy regardless of the game board structure or the opposing player’s data:A cooperative agent: The cooperative agent always implements a focal point solution. The rationale behind this policy is that the agent potentially reduces the penalties received in case of a disagreement regarding the assignment of a specific disc. On the other hand, the agent also reduces the potential profit since the agent will not try to increase individual payoffs by assigning all the discs to itself.A greedy agent: The greedy agent always assigns all the discs to itself and focuses on maximizing its profit. The motivation behind this policy is that the expectancy of the total payoffs from taking all the possible discs in the game board will be greater than the total penalties given in cases of disagreement between the players. Thus, the agent increases the chances of gaining a maximal profit but at the same time also increases the chances of receiving penalties for instances of disagreement. Another motivation behind this strategy is that the total payoffs will always be greater than or equal to that of the opponent player. Thus, the distribution of net profit associated with this agent is clearly contingent on the amount of penalties received during the game.

The performance of each of the above agents was compared against the performance of human players as well as against the performance of an agent that stochastically combines the greedy and cooperative strategies by constructing an adaptive agent as described in the next section below.

#### 2.4.2. Modeling of an Adaptive Agent

The third agent being tested was an adaptive agent that combines the approaches of the two previous agents (cooperative and greedy). The agent’s policy (either cooperative or greedy) is dependent on the probability that the opposing player will implement a focal point solution in a specific game board. The agent’s set of actions was inspired by the probabilistic technique for approximating the global optimum of a given function called “simulated annealing” (e.g., [35,36,37]). The set of actions performed by the agent in each game can be described as follows:

Calculate the opponent’s probability of implementing a focal point solution in the current game board (P_FP_) using the behavioral model.

Generate a random number X according to a uniform distribution, X~U[0, 1].Choose a specific solution for the current game by comparing between P_FP_ and X:P_FP_ > X: The adaptive agent chooses a focal point solution.P_FP_ ≤ X: The adaptive agent chooses a greedy solution.

This adaptive agent, by utilizing the behavioral model, takes advantage of the fact that it can predict the probability of implementing a focal point solution by the opposing player and act accordingly. The adaptive agent uses a stochastic policy and therefore takes advantage of both classes of agents, the cooperative and the greedy. Assuming that the behavioral model is valid, it can enable the agent to approximate global optimization while implementing the stochastic policy, thus generating a greater profit accumulated across a set of games. The model of the adaptive agent can be visualized in Figure 1.

Since the adaptive agent is stochastic (i.e., in each game iteration, the agent plays one of two classes, either cooperative or greedy), it can produce two different outcomes in a given board. Therefore, in order to evaluate the performance of the agent, its stochastic policy must be tried many times, that is, it must play many games in order to estimate its expected profit. By assuming that the number of games aspires to infinity, the average profit of the adaptive agent can be calculated based on the values of the behavioral model as follows:(3)$$E\left(profit\;\left(i,j\right)\right)=p{\left(i,j\right)}_{FP}*profit\;{\left(i,j\right)}_{FP}+(1-p{\left(i,j\right)}_{FP})*profit\;{\left(i,j\right)}_{Greedy}$$
where:*i*—the serial number of the player facing the agent*j*—the serial number of the game board*p*(*i,j*)_*FP*_—the probability that player *i* will implement a focal point solution in game board *j**profit*(*i,j*)_*FP*_—the profit of the focal agent in game board *j* against player *i**profit*(*i,j*)_*Greedy*_—the profit of the greedy agent in game board *j* against player *i*

## 3. Results

### Evaluating Agents’ Performance

Since the cooperative and greedy agents do not take into consideration the SVO of the opposing player, it was important to create an additional agent class which considers the SVO but not in a probabilistic manner. This has enabled us to estimate the performance of the adaptive agent without the bias of the prior knowledge of the SVO lacking in the case of the other two agent classes. Hence, we have created a “dichotomous” agent which employs a deterministic policy based on the SVO angle of the player. Specifically, the class of the agent (cooperative or greedy) was determined based on the cutoff point of 22.5°. This cutoff point was chosen to separate between prosocial (>22.5°) and individualistic players. Thus, whenever the SVO of the player is greater than the cutoff value, the agent chooses to consistently play cooperatively, and otherwise selfishly.

Now that we have four different agents (adaptive, dichotomous, greedy, and cooperative) that can replace the human players in the coordination games, we will evaluate the effect of each policy on the corresponding payoff. The agents will be tested as follows. In the first stage, the agent will replace the first player (the blue player, type 1). Each of the three classes of agents will play all 10 Bargaining Table games against all 52 players of type 2 (the orange players). At the end of each session (which includes 10 games), we will summarize the total points received by the agent while playing against each player and save the score. In the second stage, the agent will replace the type 2 players (the orange players) and will play the 10 Bargaining Table games against the 41 type 1 (blue) players. Recall that for the adaptive agent, the profit was calculated by using Equation (3).

Figure 2 and Figure 3 display boxplots of the distribution of payoffs accumulated within a session. Figure 2 presents the results in the case where the agents replace type 1 players and Figure 3 when they replace type 2 players. The bolded line in each figure separates between agents that base their decision on the SVO of the player (above the line) and agents that do not consider the SVO at all.

The boxplots (Figure 2 and Figure 3) reveal three important findings. First, comparing between the greedy and the cooperative agent, the former is characterized by a larger mean profit, a lower minimal profit value, and a higher maximum. This agrees with the greedy policy by which the agent assigns to itself all the discs on the board, regardless of the layout of discs, and therefore increases its potential profit to the possible maximum. However, at the same time, it takes the risk of incurring a high amount of loss in each game. In contrast, the cooperative agent assigns to itself only the points that are in agreement with the focal point solution by following the rule of closeness (i.e., Z = 1, see Appendix B). Thus, the cooperative agent reduces its potential payoff but at the same time also minimizes its potential losses.

Second, each of the three classes of agents outperformed the human players, except for the cooperative agent when replacing type 2 players. However, even in the latter case, the cooperative agent was associated with a slightly higher mean profit and a lower minimal profit value compared to the human players (Figure 3).

Third, the adaptive agent outperformed all other agent types as well as the human players and the dichotomous agent. The comparison to the latter is important since, similarly to the adaptive agent, the dichotomous agent considers the SVO of the opposing player, but independent of the probabilities predicted by the behavioral model. Because the adaptive agent’s policy is based on a weighting of the two classes of agents, cooperative and greedy, it can be inferred that the behavioral model is effective in achieving an optimal mixture of the occurrences of the two agent classes and corroborates its validity. Hence, the adaptive agent predicts when it is worthwhile playing selfishly (most of the time implementing a greedy policy) and when to cooperate with the opponent (by prioritizing cooperation).

## 4. Discussion

To the best of our knowledge, this study presents the first attempt to construct an intelligent agent that maximizes its utility by incorporating the belief system of the player in the context of tacit bargaining. The findings of this study showed that the payoff distribution obtained by the autonomous adaptive agent incorporating the SVO in the behavioral model was better than the results obtained by the human players who played against each other (i.e., the distribution associated with the agent had a higher mean value). Moreover, the payoff distribution gained by the adaptive agent was better than any of the homogeneous agent classes, namely, the greedy or the cooperative agent.

Noteworthily, the adaptive agent displayed the best performance although there was a penalty of only 20% of the disc value in cases of non-coordination. Thus, it is expected that if the penalty for non-coordination would have been higher, the difference between the adaptive and the other classes of agents would have been increased even more. This is because in a situation where the expected penalty is larger, players are expected to be more conservative and would therefore prefer to venture less frequently in comparison to the adaptive agent, which takes a calculated risk while relying on the probabilistic behavioral model.

A recent study that examined the relationship between penalty size for non-coordination and the attained payoff [23] corroborates the relationship between penalty size and risk aversion. In that study, the penalty value for non-coordination while playing the Bargaining Table game ranged between 10 and 90% and a quadratic relationship was found between penalty size and the accumulated payoff. Specifically, as the size of penalty increased, the total payoff decreased in accordance with the tendency of players to avoid potential losses [30,38]. The relatively small penalty size may also explain the relatively small difference between the greedy and the adaptive agent (see Figure 2 and Figure 3). Had the penalty size been increased, it would have been expected that the difference between these agents would be even more pronounced due to the larger accumulated cost associated with a consistent greedy policy.

It should be noted that the adaptive agent was the only agent class exposed to the SVO of the opposing players apart from the dichotomous agent (see Figure 2 and Figure 3). The fact that the adaptive agent outperformed the dichotomous agent validates the efficiency of the behavioral model since the dichotomous agent, although it considers the SVO of the opposing player, does not rely on the probabilities predicted by the behavioral model but only on the cutoff point of the SVO angle. The adaptive agent also outperformed the human players (see Figure 2 and Figure 3). This advantage of the adaptive agent may be explained by previous findings showing that there is a large range of tacit coordination ability levels among human players (e.g., [24,39,40,41]). Therefore, it is not surprising that the adaptive agent, which makes decisions that are dictated by a probabilistic model, outperforms the human players comprising strong but also weak and mediocre coordinators.

Overall, our findings can contribute to the development of agent models dealing with human–machine cooperation where collaboration is constrained by the cost of communication. In scenarios like these, the intelligent agent needs to trade off the cost of communication against its potential benefits [42]. Incorporating a behavioral model like ours into the agent model could guide the agent in deciding when it is worthwhile to communicate taking into consideration the associated cost. Our model can also aid in reinforcement learning (e.g., [43,44,45]) where the intelligent agent needs to attribute a value to a certain state. When multiagents are involved, this task is more complicated and therefore the agent can use opponent modeling to estimate the policies employed by other agents and compute the expected probabilities of the joint actions of the other agents [46]. Lastly, since our model proved to be effective in the context of tacit bargaining, it can also be applicable in contexts requiring bargaining solutions such as in human–machine networks (HMN) [19,20].

There are some limitations of the current study that warrant consideration. First, in this study we used a homogeneous sample of participants who were all students at the same university. Since it has been previously shown [24,41] that the players’ behavior in coordination games is sensitive to the effect of the cultural background, it is important to extend the study to include diverse populations.

Second, in contrast to the autonomous agent, the human players had no information regarding the SVO of the opposing players and based their decisions only on the board layout. Therefore, the comparison between the adaptive agent and the human players is somewhat biased. In future studies it would be interesting to compare between the adaptive agent and human players who will also have knowledge regarding the SVO of the opposing player. It will also be interesting to compare between human players with and without information regarding the SVO of the opposing player.

The current findings open new avenues for future research as follows. First, as mentioned earlier, we used a fixed value of the non-coordination penalty and therefore the effect of the penalty size was not examined and was not included as part of the autonomous agent’s model. This effect should be examined in future studies by varying the penalty size (see [23]) and by evaluating the performance of both the autonomous agent and the human players. Second, it will be interesting to examine how risk preferences [30] affect the performance of both the adaptive agent and the human players and compare between risk-averse and risk-seeking models and preferences. Furthermore, it would be worthwhile constructing a learning agent model so that the agent will be able to base its predictions on the SVO of the opposing player as well as on past information, that is, on its own behavior or on the behavior of the human opponent (cooperative or greedy) in previous trials (see [47]).

Third, it will be interesting to explore the relationships between electrophysiological measures and coordination ability. For example, Ref. [48] found that there is a negative relationship between the player’s individual coordination ability and the theta–beta ratio, which is a measure of cognitive load. As this latter finding was found in the context of tacit coordination with common interest, it will be worthwhile exploring this relationship in the context of divergent interests as well.

## 5. Conclusions and Future Work

In this study we constructed an intelligent agent that maximizes its utility by incorporating the belief system of the player in the context of tacit bargaining. We have shown that the SVO feature is crucial for increasing the accuracy of the predictive model. Furthermore, we have shown that an adaptive agent based on a behavioral model can outperform the dichotomous, greedy, and cooperative agent models, as well as the performance of the human players. This reward-maximizing strategy selection process based on the SVO can also be potentially applied in other human–machine contexts, including multiagent systems.

In recent years collaborative robots have become major market drivers in industry 5.0 [49], which aims to incorporate them alongside humans in a wide array of industries and applications such as assembly lines, inspection and control of operations [50,51,52], automated advising [53,54], rehabilitation, and search-and-rescue tasks [55]. In collaborative environments involving human–agent teams, sharing of cognitive elements is essential. Thus, the artificial agent is expected to adopt a human-centric strategy while attempting to perform a collaborative task and rely on shared goals derived from the human strategy to be effective in assisting the human agent in performing the joint task [6].

Hence, since industry 5.0 emphasizes the importance of optimizing human–robot collaboration, having a cognitive mechanism that mimics the reasoning process of the human agent will make the robot more robust to changes in the environment and more adaptable to different domains. Therefore, future studies should consider similar models to the one suggested here to utilize focal point solutions in scenarios where humans and robots collaborate within industrial settings.

## Figures and Tables

**Figure 1 sensors-20-07026-f001:**
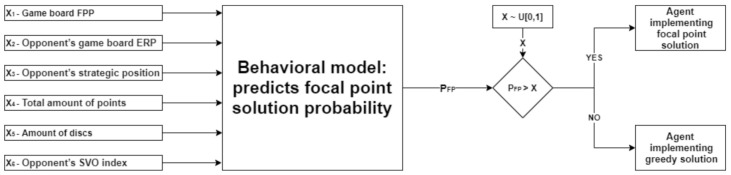
Adaptive agent—game strategy.

**Figure 2 sensors-20-07026-f002:**
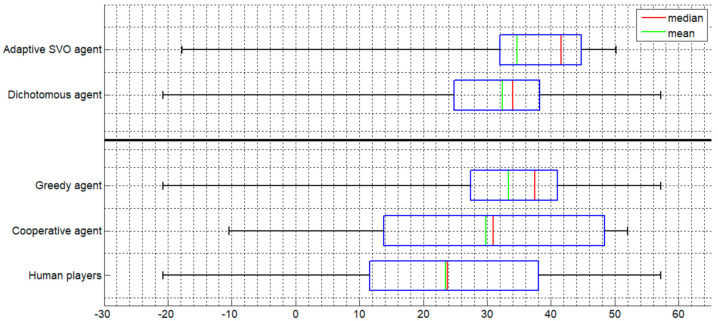
Agent plays as a type 1 (blue) player.

**Figure 3 sensors-20-07026-f003:**
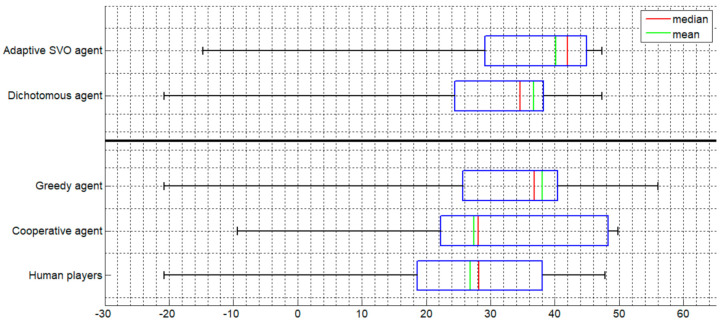
Agent plays as a type 2 (orange) player.

**Table 1 sensors-20-07026-t001:** Confusion matrix of the model with the SVO feature.

		**Predicted Classes**		**True** **Positive** **Rate**	**False** **Negative** **Rate**
		**0**	**1**		
**True** **Classes**	0	221	28	88.76%	11.24%
	1	29	102	77.86%	22.14%
**Positive Predicted Value**		88.40%	78.46%	Prediction Accuracy 85%	
**False Discovery Rate**		11.60%	21.54%		

**Table 2 sensors-20-07026-t002:** Confusion matrix of the model without the SVO feature.

		**Predicted Classes**		**True** **Positive** **Rate**	**False** **Negative** **Rate**
		**0**	**1**		
**True** **Classes**	0	179	70	71.88%	28.12%
	1	52	79	60.30%	39.70%
**Positive Predicted Value**		77.48%	53.02%	Prediction Accuracy 67.89%	
**False Discovery Rate**		22.52%	46.98%		

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
