# Peer review of "Using a Stochastic Agent Model to Optimize Performance in Divergent Interest Tacit Coordination Games"

_sensors, 2020, doi:10.3390/s20247026_

Round 1

Reviewer 1 Report

The paper addresses a relevant question: is SVO useful to construct an automated agent making choices in divergent interest tacit coordination
games?

The answer is affirmative, in that such an agent performs better than human players, and also better than automated agents always choosing a greedy or a focal point action.

I have only three relatively minor concerns:
1. SVO is often used as a measure of altruism. The focal point solution, however, may be more strongly predicted by a measure of inequity aversion, such as an estimate of the inequity-averse utility function (Fehr and Schmidt, Quarterly Journal of Economics, 1999). I invite the authors to better discuss their choice to focus on SVO, and mention that other possibilities can be explored.  
2. I would appreciate if the authors could provide further details on the procedures used to collect data. In particular, I did not understand (but it may be my fault) whether human players were matched in different pairs for each of the 10 games players, and if they were informed of the results of the already played games. This is relevant because, if the same two players repeatedly interact together, looking at the partner's play in previous rounds, then a reciprocal behavior can emerge, and this may be exploited to further improve the results of the automated agent. I invite the authors to explicitly provide this information, and discuss its potential use.
3. To fully appreciate the improvement in results due to ithe inclusion of SVO, it may be useful to present a comparison with an automated agent designed to exploit information from X1 to X5 (i.e., excluding SVO). I would leave to the authors the assessment of whether this comparison is indeed useful, and (if so) how to present it.

Author Response

Our responses to reviewer #1 comments:

We thank the reviewers for their constructive comments that helped improve the quality of our paper. Our responses to the reviewers’ comments are detailed below per each comment.

Our responses are given below each comment in bold type. The main changes in the revised manuscript are highlighted in yellow.

  1. SVO is often used as a measure of altruism. The focal point solution, however, may be more strongly predicted by a measure of inequity aversion, such as an estimate of the inequity-averse utility function (Fehr and Schmidt, Quarterly Journal of Economics, 1999). I invite the authors to better discuss their choice to focus on SVO, and mention that other possibilities can be explored.

We have addressed this point in the introduction section, page 2, lines 74-78. We have now emphasized that the motivation for the cooperative behavior may also stem from inequity-aversion, and that our agent is indifferent to the actual motivation that leads to a focal point solution.

  1. I would appreciate if the authors could provide further details on the procedures used to collect data. In particular, I did not understand (but it may be my fault) whether human players were matched in different pairs for each of the 10 games players, and if they were informed of the results of the already played games. This is relevant because, if the same two players repeatedly interact together, looking at the partner's play in previous rounds, then a reciprocal behavior can emerge, and this may be exploited to further improve the results of the automated agent. I invite the authors to explicitly provide this information, and discuss its potential use.

We thank the reviewer for raising this important point. Human players were not matched in different pairs and were not informed regarding the results of the already played games. We have now elaborated on this point on page 3, lines 126-131. We deliberately wanted to avoid dynamic changes in strategy that stem from reciprocal behavior thus players were not informed regarding the results of the previous iterations.

  1. To fully appreciate the improvement in results due to the inclusion of SVO, it may be useful to present a comparison with an automated agent designed to exploit information from X1 to X5 (i.e., excluding SVO). I would leave to the authors the assessment of whether this comparison is indeed useful, and (if so) how to present it.

We have indeed compared the importance of the inclusion of the SVO model but originally decided to leave it out of the manuscript. Excluding the SVO feature resulted in worse prediction accuracy as can be seen in the revised text (see page 5 for newly added confusion matrices), prediction accuracy drops from 85% with the SVO feature to 67.89% in the case of the model without the SVO feature.

Reviewer 2 Report

The document "Using a stochastic agent model to optimize performace in divergent interest tacit coordination games" shows four different agents named as adaptive, dichotomous, greedy and cooperative used to replace human players in coordination games. Parameters such as focal point prominence, expected revenue proportion, player's strategic position, total number of points, total number of disks and the social venue orientation were used to model players behavior. Authors state that the adaptive agent has the best performance.

Indeed, this study is quite interesting and provides several advances in the field of agents. In general, the paper is easy to read and follow, it has a correct structure, with proper english language; however, before I emit a favorable decision there are some issues that can be improve in the paper.

Abstract: please define SVO. 

It is strange to intruduce industry 5.0, and collaborative robots when the paper deals with coordination games. In fact I was sure that the document somewhere will deal with collaborative robots, however, this never happens.

Results section is quite short, is there any additional results to be included, i.e., those shown in appendix, including the table?

Discussion, clarify the possible application in Industry 5.0

Conclusion: although it is not mandatory I suggest to include a conclusion section, including the future approaches.

Author Response

Our responses to reviewer #2 comments:

We thank the reviewers for their constructive comments that helped improve the quality of our paper. Our responses to the reviewers’ comments are detailed below per each comment.

Our responses are given below each comment in bold type. The main changes in the revised manuscript are highlighted in yellow.

  1. Abstract: please define SVO.

Corrected in abstract.

  1. It is strange to intruduce industry 5.0, and collaborative robots when the paper deals with coordination games. In fact I was sure that the document somewhere will deal with collaborative robots, however, this never happens.

We have now edited the manuscript to accommodate this comment. Specifically, we have moved the first paragraph of the introduction to a new paragraph at the end of the manuscript entitled “Future work” (on page 10). Furthermore, we have elaborated the original paragraph and explained the relevance of industry 5.0 to our study.

  1. Results section is quite short, is there any additional results to be included, i.e., those shown in appendix, including the table?

Yes, we have added two new tables in order to emphasize the importance of the inclusion of the SVO feature into the model. Please see the newly added tables and the related description to the results section on page 5.

  1. Discussion, clarify the possible application in Industry 5.0

Please see our response to the above comment #2.

  1. Conclusion: although it is not mandatory I suggest to include a conclusion section, including the future approaches.

We have now added a conclusion and future work section at the end of the manuscript on page 10.

Round 2

Reviewer 2 Report

All my concerns have been fullfilled by authors. No more comments. Congratulations